# α7 nicotinic acetylcholine receptor interaction with G proteins in breast cancer cell proliferation, motility, and calcium signaling

Murat Oz[1]☯, Justin R. King[2]☯, Keun-Hang Susan Yang[3], Sarah Khushaish[1], Yulia Tchugunova[1], Maitham A. Khajah[1], Yunus A. Luqmani[1], Nadine Kabbani[2,4]*

1 Department of Pharmacology and Therapeutics, College of Pharmacy, Kuwait University, Safat, Kuwait, 2 Interdisciplinary Program in Neuroscience, George Mason University, Fairfax, Virginia, United States of America, 3 Department of Biological Sciences, Schmid College of Science and Technology, Chapman University, Orange, California, United States of America, 4 School of Systems Biology George Mason University, Fairfax, Virginia, United States of America

☯ These authors contributed equally to this work.
* nkabbani@gmu.edu

**Data Availability Statement:** All relevant data are within the paper and its Supporting information files.

## Abstract

Chronic smoking is a primary risk factor for breast cancer due to the presence of various toxins and carcinogens within tobacco products. Nicotine is the primary addictive component of tobacco products and has been shown to promote breast cancer cell proliferation and metastases. Nicotine activates nicotinic acetylcholine receptors (nAChRs) that are expressed in cancer cell lines. Here, we examine the role of the α7 nAChR in coupling to heterotrimeric G proteins within breast cancer MCF-7 cells. Pharmacological activation of the α7 nAChR using choline or nicotine was found to increase proliferation, motility, and calcium signaling in MCF-7 cells. This effect of α7 nAChR on cell proliferation was abolished by application of Gαi/o and Gαq protein blockers. Specifically, application of the Gαi/o inhibitor pertussis toxin was found to abolish choline-mediated cell proliferation and intracellular calcium transient response. These findings were corroborated by expression of a G protein binding dominant negative nAChR subunit (α7$_{345-348A}$), which resulted in significantly attenuating calcium signaling and cellular proliferation in response to choline. Our study shows a new role for G protein signaling in the mechanism of α7 nAChR-associated breast cancer growth.

## Introduction

Epidemiological studies show that smoking of tobacco products can significantly increase a woman's risk of developing breast cancer [1–4]. Thus, although cigarette smoke consists of a complex mixture of > 3000 chemicals, nicotine is a primary bioactive component of tobacco contributing to cancer risk [5]. A large body of published work relates nicotine exposure and the proliferation as well as metastatic potential of cancer cells within tissue such as lung and breast [6]. Molecular studies show that nicotine increases the proliferation of cancer cells through the activation of nicotinic acetylcholine receptors (nAChRs) expressed at the cell

**Funding:** Funding provided by the Kuwait Foundation for the Advancement of Sciences (PN19-13PT-01). The funders had no role in study design, data collection and analysis, decision to publish, or preparation of the manuscript.

**Competing interests:** The authors have declared that no competing interests exist.

surface [7]. In some cases, nAChR expression can be altered during cancer cell growth and through hormones or toxin exposure [8].

The nAChR channel is a pentameric protein consisting of combinations of 16 isoforms of subunits including a subunits $\alpha 1$–$\alpha 10$, $\beta 1$–$\beta 4$, $\gamma$, $\delta$, and/or $\varepsilon$ [9, 10]. Homopentameric nAChRs including $\alpha 7$ and $\alpha 9$ exhibit high calcium permeability and thus can directly signal long-term cellular growth [11, 12]. Indeed, nicotine has been shown to activate $\alpha 7$ [13–17] and $\alpha 9$ [18–21] nAChRs in breast cancer cells. In cooperation with calcium influx, ligand activation of $\alpha 7$ nAChRs drives metabotropic signaling cascades important for growth and cytoskeletal motility [22–25]. In this study, we explored the role $\alpha 7$ nAChR interaction with G proteins in MCF-7 breast cancer cell proliferation, motility, and calcium signaling.

## Materials and methods

### Cell culture and transfection

Michigan Cancer Foundation-7 (MCF-7) cell line (ATCC, Manassas, VA, USA) cells was maintained as monolayers in advanced Dulbecco's minimum essential medium (DMEM) containing phenol red and supplemented with 5% fetal bovine serum (FBS), 600 μg/ml L-glutamine, 100 U/ml penicillin, 100 μg/ml streptomycin and 6 ml/500 ml 100 x non-essential amino acids (all from Invitrogen, CA, USA), and grown at 37˚C in an incubator of 5% $CO_2$ and 95% humidity. MCF-7 is a human breast cancer cell line with estrogen, progesterone and glucocorticoid receptors and is derived from the pleural effusion of a 69-year-old metastatic adenocarcinoma. MCF-7 cells were transfected with constructs encoding human $\alpha 7_{345\text{-}348A}$ or the control human $\alpha 7$ nAChR in pEYFP-C1 (Addgene) [24], GCaMP5G [26] using Lipofectamine 2000 (Thermo Fisher, Waltham, MA, USA). DNA was purified using a maxi prep kit (Xymo Research, Irvine, CA, USA).

### Drug preparation

Choline, nicotine, cotinine, nornicotine, methyllycaconitine citrate (MLA) and $\alpha$-bungarotoxin (BTX) were from Sigma Aldrich (St. Louis, MO, USA). Choline, MLA, and BTX were dissolved in distilled water. Nicotine, cotinine, and nornicotine were dissolved in ethanol. YM 254890 and pertussis toxin (PTX) were purchased from Tocris-Bio-Techne Corporation (Minneapolis, MN, USA) and dissolved in DMSO and distilled water, respectively. $\alpha$-Conotoxin RgIA (CTX) was obtained from Alomone Labs (Jerusalem, Israel) and dissolved in distilled water.

### Cellular fluorescence

Cell were permeabilized with 0.05% Triton X-100 then blocked with 10% goat serum (Life Technologies, Carlsbad, CA, USA) [27]. Surface $\alpha 7$ nAChRs were visualized in non-permeabilized cells using 100 nM Alexa Fluor (488 or 647) conjugated BTX as described [28, 29]. Imaging was performed on an inverted Zeiss LSM800 confocal microscope (Carl Zeiss Oberkochen, Germany). Analysis was carried out in ImageJ (NIH, Bethesda, MD, USA).

### Cell proliferation (MTT assay)

Approximately $10^4$ cells were seeded into triplicate wells of 12-well plates and allowed to attach overnight. Growth was assessed by an MTT assay after 3 days of drug incubation. Briefly, 1 ml of MTT [3-(4,5-dimethyl thiazolyl-2)-2,5-diphenyltetrazolium bromide] reagent (Promega) (0.5 mg/ml) was added to each well and incubated at 37˚C for 30 min before the addition of 1 ml acidic isopropanol then vigorous re-suspension of the converted blue crystals. Cells were

optically counted with a hemocytometer (Thermo Fisher) or the absorbance of the suspension was measured at 595 nm with background subtraction at 650 nm. Results are expressed as mean ± standard error of the mean (S.E.M.).

## Calcium imaging

Cells were transfected with GCaMP5G [26] and the signal was detected using a Zeiss LSM 800 at an acquisition rate of 1 frame per 256 ms for 75 sec at 2 x 2 binning as described previously [23]. Drugs were applied to the recording chamber via a gravity fed perfusion at a flow rate of 1 ml/sec. Regions of interest (ROIs) were normalized as ΔF/Fθ and analyzed using ImageJ (NIH) as described [23]. A total of 20–30 cells were imaged per experimental condition and experiments were performed in triplicate.

## Cell motility assay

Cells were plated on 6-well plates at 80–90% confluence with complete DMEM containing vehicle or drug. The following day, a scratch was created in the cell monolayer using a sterile p1000 pipette tip. A photograph of the scratched area was taken immediately (0 h) and after a 24 h incubation in a 37°C, 5% $CO_2$. Cell motility was determined by calculating the width of the scratch at 24 h as a percentage of 0 h.

## Statistical analysis

Student's two-tailed unpaired t-test, or one-way ANOVA test followed by Bonferroni post hoc test were used to compare means of individual groups with $p < 0.05$ as statistically significant.

# Results

## Nicotinic receptor ligands promote proliferation and motility in MCF-7 cells

Nicotine has been shown promote cancer cell proliferation within various cell lines and in animal systems [6, 7, 20]. We examined the effect of a 3-day treatment with nicotine (50 nM-1 μM) on MCF-7 cell proliferation (Fig 1A). Treatment with nicotine was found to produce a concentration dependent increase in cell proliferation (p<0.05; ANOVA). Similarly, treatment with the selective α7-nACh receptor agonist choline increased proliferation of MCF-7 cells (Fig 1B). Cotinine, is the primary metabolite of nicotine and can persist longer in the body that nicotine [30]. Nornicotine, like nicotine, is found in tobacco products and has carcinogenic properties [31]. Both cotinine and nornicotine activate mammalian nAChRs and contribute to addiction [32]. We tested the effect of a 3-day treatment with cotinine (0.1–10 μM) or nornicotine (0.1–10 μM) on cell proliferation. We found that both cotinine and nornicotine increased cell proliferation to levels comparable with choline and nicotine at the concentration range of 1–10 μM (Fig 1C and 1D).

Nicotine can activate calcium signaling and lead to cell proliferation and the regulation of apoptosis within cancer cells [20]. We investigated the involvement of α7 and α9-nAChRs in breast cancer proliferation in response to 10 mM choline application, which activates both homopentameric receptor types [33]. Co-application of α7-nAChR antagonists: methyllycaconitine (10 μM, MLA) or α-bungarotoxin (100 nM, BTX), was found to significantly reduce cell proliferation in this experiment. Conotoxin RgIA (100 nM, CTX), an antagonist with higher potency for α9 and α10-nAChRs [34], also inhibited choline mediated proliferation (Fig 2).

The wound healing (or scratch) assay can measure cancer cell migration in vitro [35]. Based on earlier studies that show that α7 nAChR activation can modify the cytoskeleton and cell

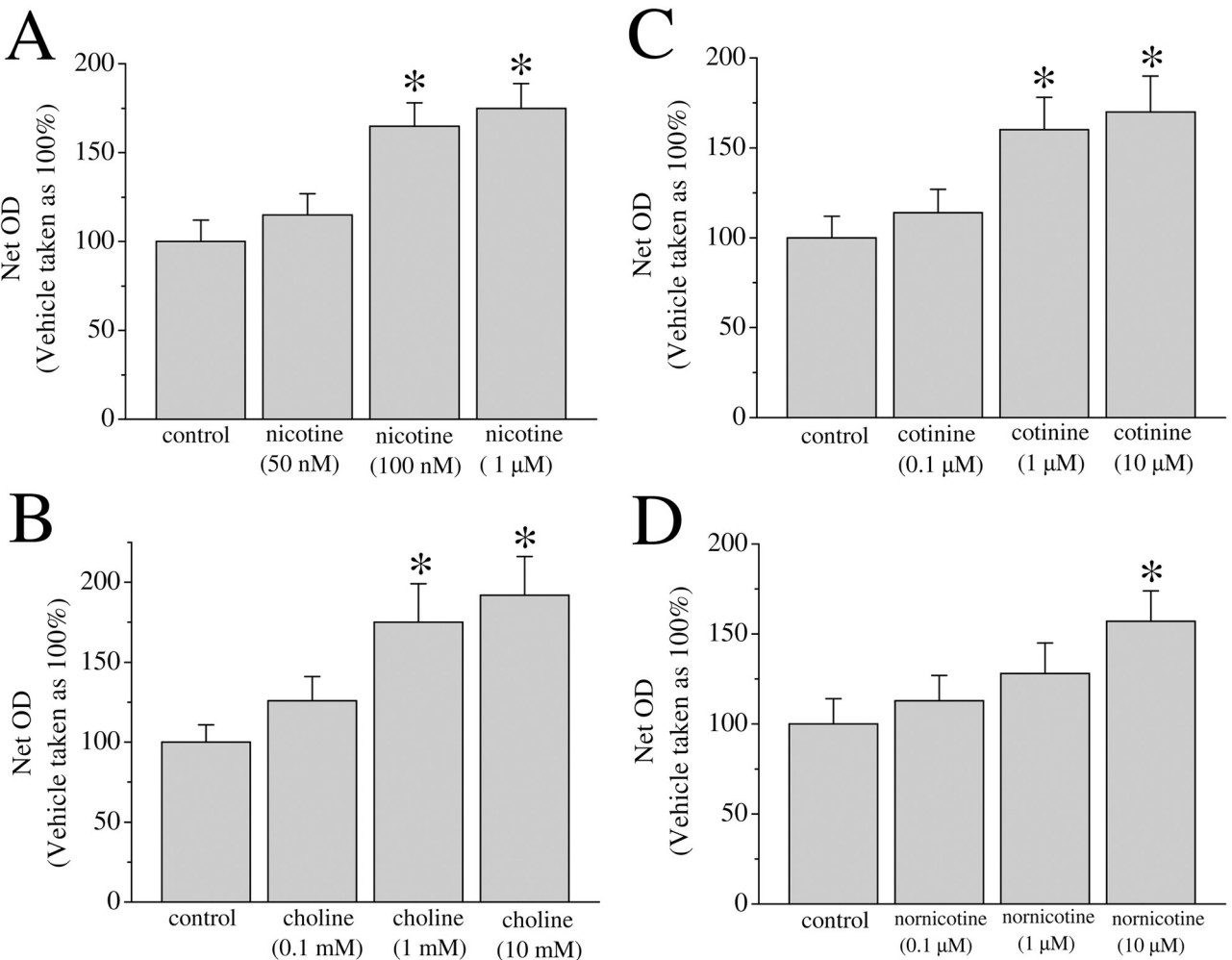

**Fig 1. Dose-response effect of nicotine, choline, and nicotine metabolites on breast cancer cell proliferation.** Approximately $10^4$ MCF-7 cells were seeded into microwell plates and grown for 3 days in the presence of vehicle (control) or increasing concentrations of nicotine (**A**), choline (**B**), cotinine (**C**), and nornicotine (**D**). Cells were harvested and growth was determined by an MTT assay. Bars represent means ± SEM of at least 3 independent determinations. Asterisk denotes significant difference from control with $p < 0.05$.

motility as well as structural change [23, 36], we tested the effect of choline on the motility of MCF-7 cells. A 24-hour treatment with 10mM choline was found increase cell motility by 2-fold when compared to the control. This effect of choline on motility was virtually eliminated by co-application of 100 nM BTX (Fig 3).

## G protein coupling to α7 nAChR drives intracellular calcium signaling and cell proliferation

Ligand stimulation of α7 nAChRs impacts receptor synthesis and trafficking to the cell surface in cancer cells [37]. We examined α7 nAChR expression at the cell surface after a 3-day treatment with 10 mM choline. In this experiment, we used a BTX-Alexa Fluor 488 conjugate to label non-permeabilized MCF-7 cells (S1 Fig), comparing choline treated to control cells. As shown in Fig 4, treatment with choline was associated with an increase in the BTX-Alexa Fluor 488 signal at the cell surface in comparison to the control condition.

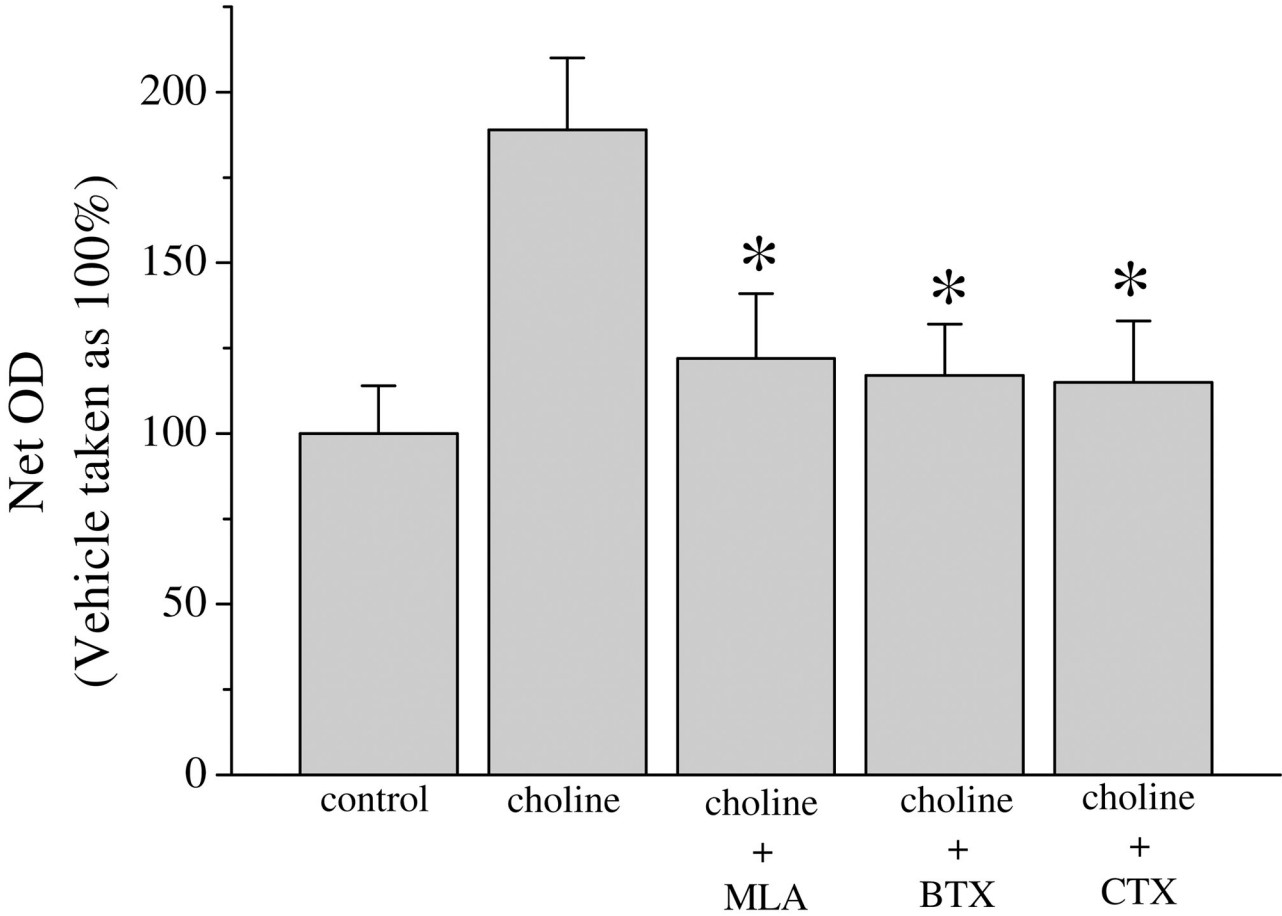

**Fig 2. Choline effect on proliferation in MCF-7 cells.** Approximately $10^4$ MCF-7 cells were seeded into microwell plates and grown for 3 days in the presence of vehicle (control), choline (10 mM), choline + methyllycaconitine (10 μM, MLA), choline + α-bungarotoxin (100 nM, BTX), and choline + α-Conotoxin RgIA (100 nM, CTX). Cells were harvested and growth was determined by an MTT assay. Bars represent means ± SEM of at least 3 independent determinations. Asterisk denotes significant difference from choline treated group with $p < 0.05$.

Ligand stimulation of the α7 nAChR rapidly increases intracellular calcium levels by extracellular calcium influx through open nAChRs, the activation of voltage-gated calcium channels, and the activation of calcium induced calcium release (CICR) as well as inositol induced calcium release (IICR) from the ER [22]. We examined the effect of 10 mM choline on intracellular calcium within MCF-7 cells. As shown in Fig 5A, application of choline resulted in a calcium transient that appeared within 0.5 sec of drug application and lasted for ~1 second. This calcium transient was not seen when BTX was present in the application solution. Statistical analysis of calcium transient peaks between drug treatment groups confirms that BTX abolishes choline-associated calcium responses within MCF-7 cells (Fig 5B).

In previous studies we have shown that G protein interaction is involved in α7 nAChR-mediated IICR [22]. We tested the effect of the Gαi/o inhibitor PTX on choline-mediated calcium transients. Co-application of PTX was found to reduce the choline calcium transient by over 30% an effect that was found to be statistically significant (Fig 5B). We further explored G protein activity in nAChR-mediated cell growth. The 3-day proliferation assay was repeated in MCF-7 cells treated with 10 mM choline alone, choline with PTX (5 μg/ml), or choline with the Gαq inhibitor YM 254890 (1 μM). Analysis indicates that application of PTX or YM

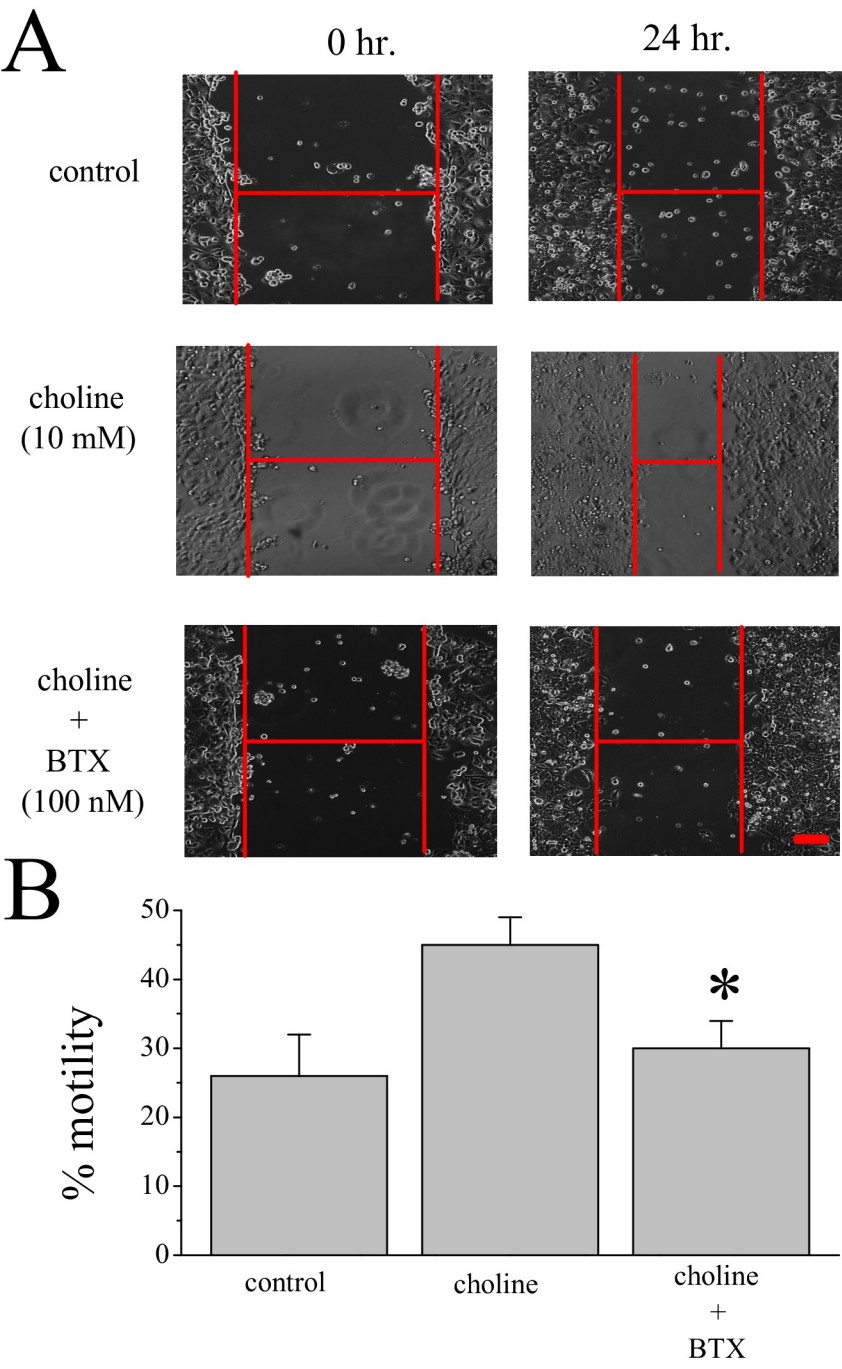

**Fig 3. Choline increases cell motility. (A)** Images of MCF-7 cells at the start (0 hr) and end (24 hr) of the motility assay. Experimental groups: vehicle (control), choline (10 mM), and choline with α-bungarotoxin (100 nM, BTX). **(B)** Average motility measures at 24 hr from 3 independent experiments. Asterisk denotes significance $p < 0.05$. Scale = 100μm.

254890 indicates a significant reduction in choline-mediated proliferation when G protein blockers are present (Fig 6).

We have shown that α7 nAChRs directly bind G proteins through a binding sequence that is located within the intracellular loop region of the receptor [24]. Site directed mutagenesis of

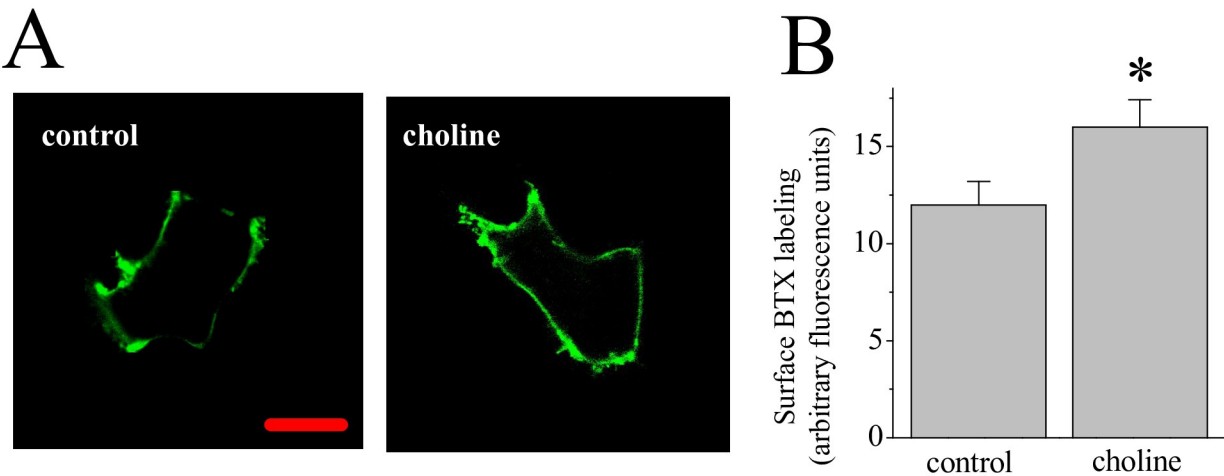

**Fig 4. An effect of choline on surface labeling for the nAChR. (A)** MCF-7 cells were incubated with 100 nM α-bungarotoxin (BTX) Alexa Fluor 488. Images of representative cells showing labeling at 72 hr in control and choline (10 mM) treated cells. Scale = 5μm (**B**) Average BTX fluorescence values. Asterisk denotes significance p < 0.05 (n = 24–27).

4 amino acids at this site creates a dominant negative α7 subunit ($α7_{345-348A}$) that is unable to bind G proteins [24]. In previous work, the expression of $α7_{345-348A}$ is sufficient to impair wild-type α7 nAChR calcium signaling and cell growth [36]. We transfected MCF-7 cells with $α7_{345-348A}$ and as shown in Fig 7, the expression of this mutant was seen sufficient in abolishing calcium transient responses to choline application. We also examined the effect of $α7_{345-348A}$ on proliferation in MCF-7 cells. An analysis of $α7_{345-348A}$ expression indicates that this mutant subunit does not significantly alter MCF-7 growth in the absence of a cholinergic ligand (Fig 7B). However, in cells transfected with $α7_{345-348A}$, a 3-day treatment with choline did not increase proliferation as evidenced by a comparison of choline effect on cell number between wild-type α7 and $α7_{345-348A}$ expressing cells (Fig 7B and 7C).

## Discussion

Evidence indicates a link between tobacco product use and increased risk to oral, lung, and breast cancer [38]. While there maybe diverse mechanisms by which nicotine can promote cancer cell progression, they all involve the activation of nAChRs on target cells [7, 20, 39]. Indeed, various nAChR couple to cancer processes that regulate cell division, morphology, and can increase angiogenesis and modify inflammatory responses in microenvironment of the cancer cells [7, 25, 40]. Recently, nAChRs have been shown to promote MCF-7 breast cancer cell proliferation via the activation of ERK1/2 phosphorylation [14] and drive epithelial to mesenchymal transition (EMT) [17].

In various neuronal cells high calcium signaling through the homopentameric α7 nAChR directs actin associated cytoskeletal dynamics leading to observable changes synaptic growth and plasticity [25, 41]. An analogous mechanism for α7 nAChR-calcium signaling to the cytoskeleton exists in non-neuronal cells including immune and epithelial cells [42, 43]. In this case, nAChR activation may drive changes in cell proliferation and/or metastatic transition involving the regulation of the cytoskeleton in cancer cells [44]. Our observations suggest that cholinergic ligands (e.g., nicotine) can alter α7 nAChR expression at the cell membrane of breast cancer cells. This may explain aspects of chronic nicotine exposure on cancer risk in smokers and suggests that trafficking of nAChRs may contribute to cancer progression.

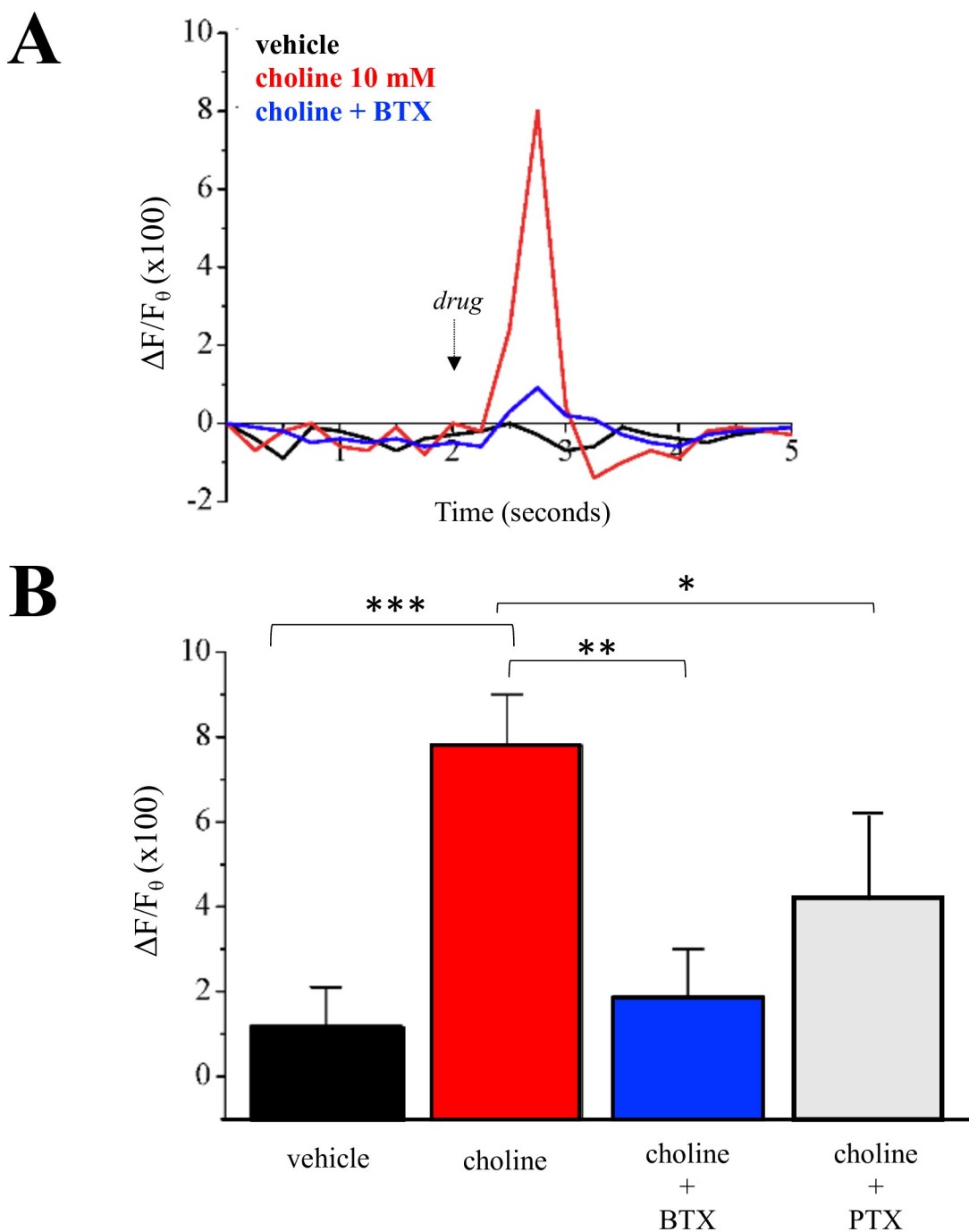

**Fig 5. Choline mediated change in intracellular calcium in MCF-7 cells. (A)** Calcium transients ($\Delta F/F\theta$) measured using GCaMP5G in response to drug application. Vehicle (control), choline (10 mM), and choline with $\alpha$-bungarotoxin (100 nM, BTX). **(B)** Average calcium peaks across experimental groups. Choline with pertussis toxin (5 µg/ml, PTX) (n = 21–25 cells). Statistical significance: *$p<0.05$; **$p<0.01$, ***$p<0.001$.

The metabotropic activity of the $\alpha$7 nAChR is driven by protein interactions of the receptor's intracellular (M3-M4) loop [45]. We have shown the existence of a G protein binding region within the $\alpha$7 nAChR M3-M4 loop [24]. Coupling between the nAChR and various G proteins, including G$\alpha$i and G$\alpha$q, was found to participate in important aspects of nAChR

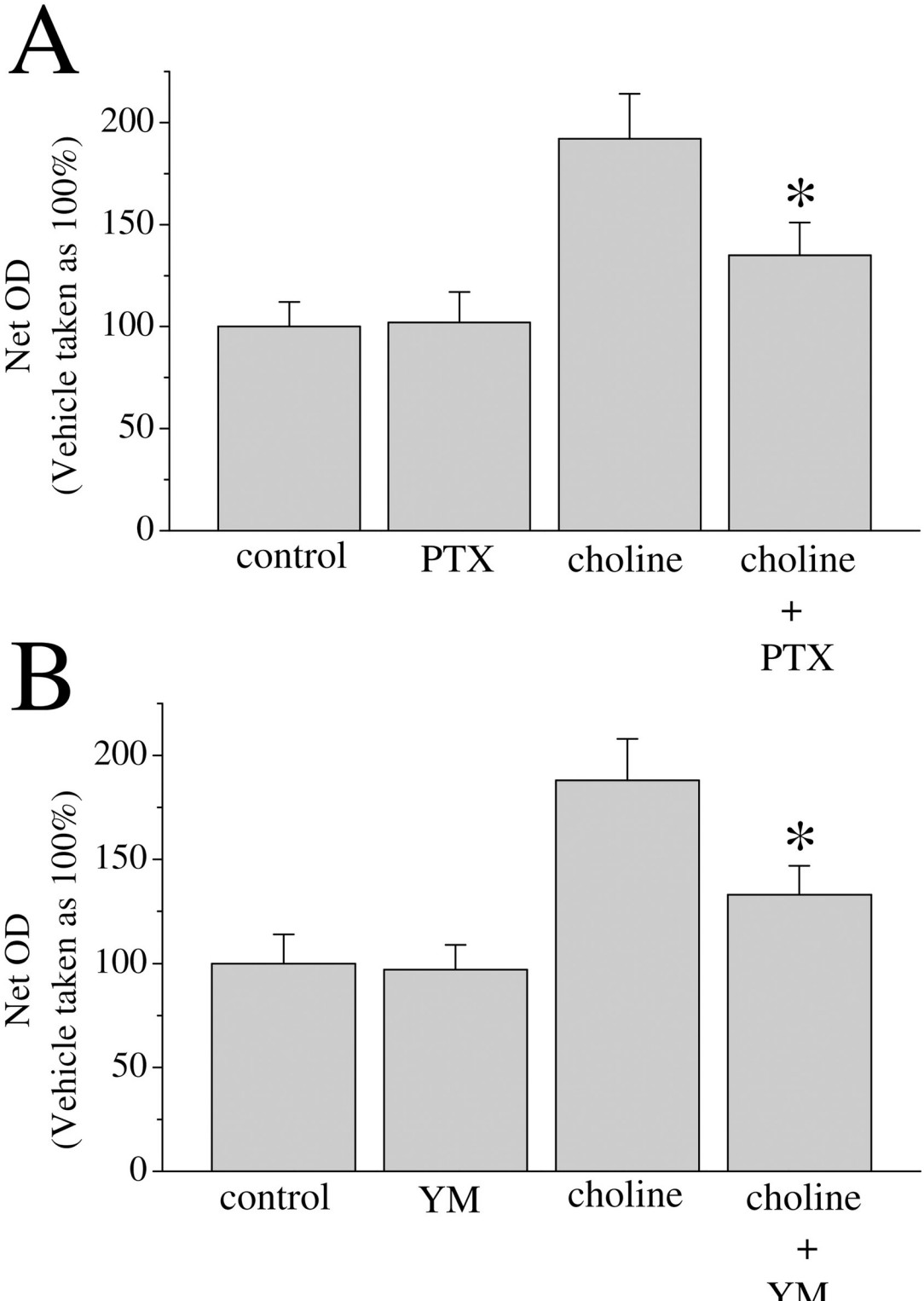

**Fig 6. G-protein inhibitors block choline-mediated MCF-7 cell proliferation.** Approximately $10^4$ MCF-7 cells were seeded into microwell plates for 3 days then assessed via an MTT assay: **(A)** vehicle (control), pertussis toxin (5 μg/ml, PTX), choline (10 mM), and choline with PTX, **(B)** vehicle (control), YM 254890 (1 μM), choline (10 mM), and choline with YM 254890. Bars represent means ± SEM of 3 independent experiments. Asterisk denotes significant difference from choline treated group ($p < 0.05$).

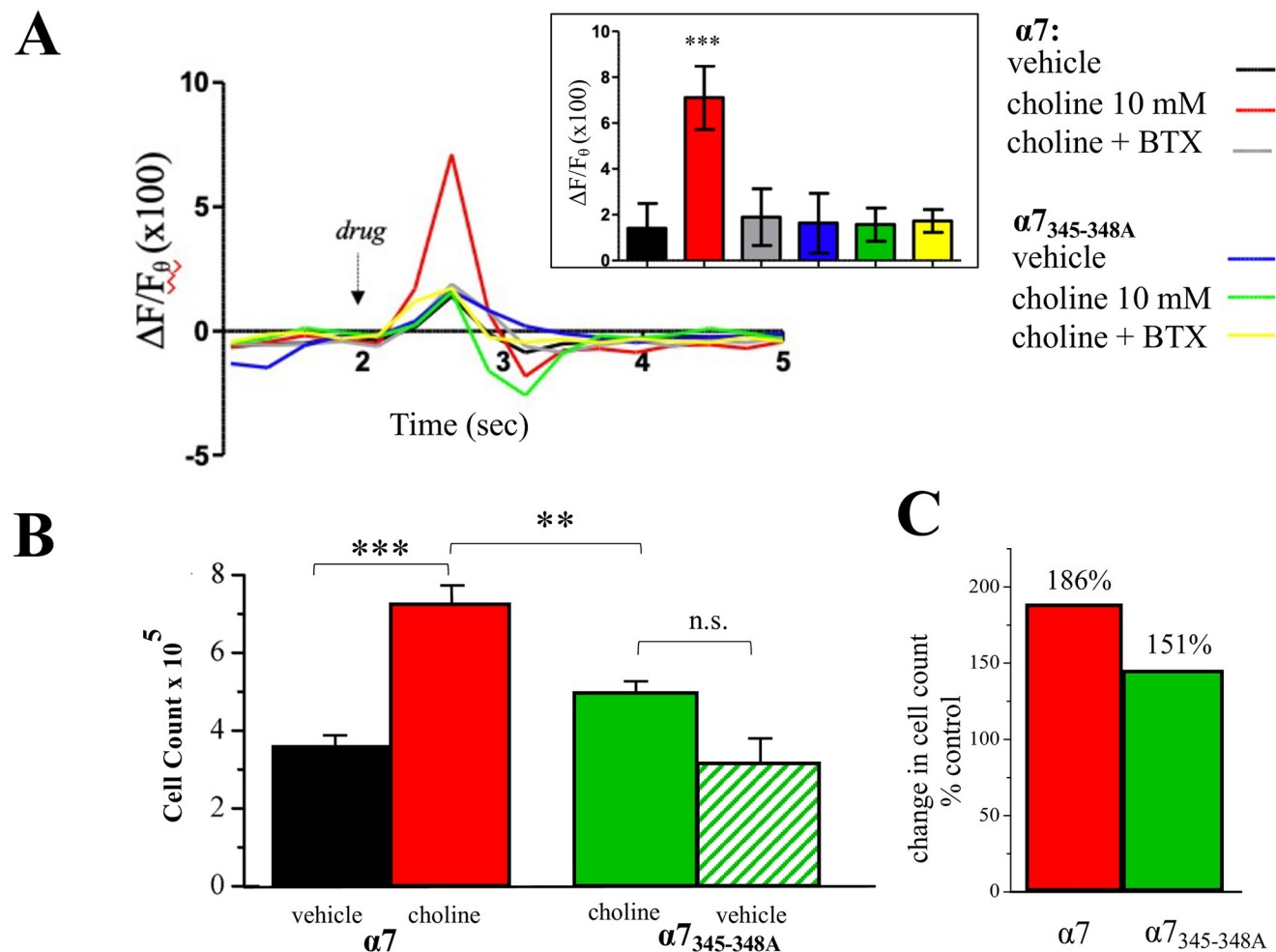

**Fig 7. G protein regulation of α7 nAChR calcium signaling. (A)** Representative calcium transients following application of vehicle (control), choline (10 mM), and choline with α-bungarotoxin (100 nM, BTX) in cells transfected with wild-type α7 or the mutant $\alpha7_{345-348A}$ subunit. Inset: average of calcium transient peak values ($\Delta F/F\theta$) across experimental groups (n = 21–25). **(B)** Approximately $10^4$ MCF-7 cells were seeded into microwell plates for 3 days then assessed via the MTT assay: vehicle (control) or choline (10 mM) in cells transfected with wild-type α7 or the mutant $\alpha7_{345-348A}$ subunit. **(C)** A comparison of choline mediated proliferation in cells transfected with wild-type α7 or the mutant $\alpha7_{345-348A}$ subunit relative to the vehicle control. Statistical significance **$p < 0.01$, ***$p < 0.001$.

signaling [25]. At present it is not clear if this G protein binding site within the α7 nAChR favors association with specific Gα subunits. Based on *in vivo* analysis of G protein interactions with the α7 nAChR in the adult rodent brain, various G protein subunits appear able to bind the nAChR [24].

In this study, functional interaction between the α7 nAChR and Gαi appears to contribute to MCF-7 cell proliferation and motility based on experiments that show that blocking Gαi/o activity with PTX can abolish choline-associated cell proliferation. In calcium imaging experiments however PTX was found to significantly reduce the choline calcium transient response but not completely abolish it thus suggesting additional (non-Gαi) contributions to the α7 nAChR calcium signal. This is consistent with experiments using the Gαq inhibitor YM 254890 showing an effect of the G protein subunit on calcium mediated MCF-7 growth. Our results are supportive of earlier finding that show that Gαq can bind the α7 nAChR and promote calcium store release [23] and Gαi can activate pathways important for breast cancer growth [46]

Mammalian α7, α9, and heteromeric α9α10 combinations are the nAChR types with the greatest $Ca^{2+}$ permeability [11, 47, 48]. It is important to note that many cancer cells express homopentameric nAChRs that are found to stimulate cancer cell proliferation, metastasis, and inhibit cancer cell apoptosis [7, 20]. The α9 nAChR contributes to breast cancer growth and EMT through activation of PI3K or MAPK signaling pathways [49]. In this study we also find a significant effect of α-Conotoxin RgIA, that is known to have a higher potency for α9 than α7 nAChRs, on MCF-7 cell growth. Thus, in future studies it will be important to explore the involvement of G proteins in α9 nAChR signaling in breast cancer cells.

## Supporting information

**S1 Fig. Cell surface labeling of nAChRs using BTX.** MCF-7 cells were labeled with 100 nM BTX. Cell membranes were not permeabilized in these labeling experiments. Bottom panel shows a representative image of a labeled cell. Scale bar = 5μm.
(TIFF)

## Author Contributions

**Conceptualization:** Murat Oz, Yulia Tchugunova, Nadine Kabbani.

**Data curation:** Justin R. King, Sarah Khushaish, Yulia Tchugunova, Nadine Kabbani.

**Formal analysis:** Murat Oz, Justin R. King, Sarah Khushaish, Yulia Tchugunova, Yunus A. Luqmani, Nadine Kabbani.

**Funding acquisition:** Murat Oz, Maitham A. Khajah, Nadine Kabbani.

**Investigation:** Nadine Kabbani.

**Methodology:** Justin R. King, Sarah Khushaish, Yulia Tchugunova, Maitham A. Khajah, Yunus A. Luqmani, Nadine Kabbani.

**Project administration:** Nadine Kabbani.

**Resources:** Murat Oz, Keun-Hang Susan Yang, Nadine Kabbani.

**Supervision:** Murat Oz, Nadine Kabbani.

**Writing – original draft:** Murat Oz, Nadine Kabbani.

**Writing – review & editing:** Murat Oz, Maitham A. Khajah, Yunus A. Luqmani, Nadine Kabbani.

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
