## [Decision Letter · Decision Letter 0]

7 Jun 2023

PONE-D-23-15038α7 nicotinic acetylcholine receptor interaction with G proteins mediates breast cancer cell proliferation, motility, and calcium signalingPLOS ONE

Dear Dr. Kabbani,

Thank you for submitting your manuscript to PLOS ONE. After careful consideration, we feel that it has merit but does not fully meet PLOS ONE’s publication criteria as it currently stands. Therefore, we invite you to submit a revised version of the manuscript that addresses the points raised during the review process. Please submit your revised manuscript by Jul 22 2023 11:59PM. If you will need more time than this to complete your revisions, please reply to this message or contact the journal office at plosone@plos.org. Please include the following items when submitting your revised manuscript:A rebuttal letter that responds to each point raised by the academic editor and reviewer(s). You should upload this letter as a separate file labeled 'Response to Reviewers'.A marked-up copy of your manuscript that highlights changes made to the original version. You should upload this as a separate file labeled 'Revised Manuscript with Track Changes'.An unmarked version of your revised paper without tracked changes. You should upload this as a separate file labeled 'Manuscript'.

We look forward to receiving your revised manuscript.

Kind regards,

Israel Silman

Academic Editor

PLOS ONE

Journal Requirements:

   "The authors acknowledge funding provided by the Kuwait Foundation for the Advancement of Sciences (PN19-13PT-01)."

Additional Editor Comments:

In your revised manuscript you are requested to address as fully as possible the comments of the two reviewers, especially the detailed and constructive comments and criticisms of Reviewer 1.

Reviewers' comments:

Reviewer's Responses to Questions

**Comments to the Author**

1. Is the manuscript technically sound, and do the data support the conclusions?

Reviewer #1: Partly

Reviewer #2: Yes

2. Has the statistical analysis been performed appropriately and rigorously? 

Reviewer #1: I Don't Know

Reviewer #2: Yes

3. Have the authors made all data underlying the findings in their manuscript fully available?

Reviewer #1: Yes

Reviewer #2: No

4. Is the manuscript presented in an intelligible fashion and written in standard English?

Reviewer #1: Yes

Reviewer #2: Yes

5. Review Comments to the Author

Reviewer #1: The experiment in figure 7B should be repeated with a control group for α7345-348a that was not exposed to choline to show that transfection with α7345-348a does not alter proliferation by itself.

The results in Figure 4 are not mentioned in the abstract or the discussion. It would be good if the authors discuss how the increase in α7 nAChR expression might affect the proliferation, motility, and/or calcium signaling of cancer cells. Additionally, the authors did not show if these effects could be blocked by MLA or another antagonist, or whether this upregulation is related to g-protein mediated signaling via α7 nAChRs. Further experiments to answer these questions would help to tie these results into the rest of the paper.

YM 254890 is mentioned in the abstract as a G protein blocker, but in the results section of the manuscript, the authors never define what specifically YM 254890 is or why it is being used.

The proliferation assay in Figure 7 plots Cell count x 105, whereas the other proliferation assays measure Net OD. Could the authors please explain why this assay was quantified differently?

There are two “B” panels in figure 5

The deltaF/F scales are not consistent across figures. For example, in Figure 5, A and C plot deltaF/F (x100), whereas panel B1 writes out the 100s, and the second panel B just has deltaF/F without the x100 clarification. Similarly, in Figure 7, panel A, the first graph is x100, while the second graph is not.

Scale bars should be added to Figure 3A and Figure 4A

Scale bar should be defined in Figure 7B.

In the figure legend for Figure 6 it is noted that the “Asterix denotes significant difference from control,” whereas the figure seems to indicate that the Asterix represents significant difference from the choline treated group. Could the authors clarify what comparisons are being measured in this figure, and what is significantly different from what?

The figure legend for Figure 2 states “Asterix denoted significant difference from control” but the figure seems to indicate that significance is measured as difference from the choline group.

Figure 7 has only 2 panels, A and B, but the figure legend has A, B, and C.

Figure 7 states “statistical significance relative to the control or α7 transfected group.” This is confusing. Could the authors clarify what comparisons are being made and what significance is being denoted. Does this mean that the α7345-348a group is significantly different from both the control and α7 group? If so, this should be denoted by two different symbols.

In the text, the mutant α7 is called “α7345-348a” whereas in the figure legend and figure it is “α7345-8a”, this nomenclature should be consistent throughout the paper

In the figure legend for Figure 5, the definition of a double Asterix is given, but there is no double Asterix in Figure 5

Reviewer #2: The paper by Oz et al, shows that in MCF-7 cells activation by choline of α7 receptor increases

proliferation, motility, and calcium signaling and this is partially blocked by the application of G protein blocker, Gαi/o inhibitor and by the presence of a G protein binding dominant negative α7 subunit (α7345-348A).

The work is interesting and well done, and it demonstrated that the α7 subtype is involved in mediating choline induced proliferation, motility, and calcium signalling. However the effect of choline was blocked using antagonists that act on both α7 and α9 subtypes. Why the authors did not used the α7 selective antagonist AR (Whitaker et al. 2007) and the α9 selective antagonist RGIA4 (Romero et al 2017) ? This certainly will help in clarifying the subtypes involved and their contribution to the effects reported

Minor points:

In the abstract it is written "Nicotine activates various nicotinic acetylcholine receptors (nAChRs)" and in the introduction it is written"Mammalian nAChRs assemble from α (α1–α6) and β (β2–β4) subunit combinations forming homo- or hetero-pentameric channels [9, 10].

Nicotine does not activates a1 containing receptors and nicotine activates various nAChRs

that assemble from α (α2–α7, α9, α10) and β (β2–β4) subunit combinations.

Fig 7: Please add C to the figure

6. PLOS authors have the option to publish the peer review history of their article (what does this mean?). If published, this will include your full peer review and any attached files.

Reviewer #1: No

Reviewer #2: No

---

## [Author Response · Author response to Decision Letter 0]

7 Jul 2023

Please find the revision of our manuscript entitled “α7 nicotinic acetylcholine receptor interaction with G proteins mediates breast cancer cell proliferation, motility, and calcium signaling” for consideration as a research article in PLOS One. 

We have taken into consideration the comments of both reviewers and have addressed them throughout the revised manuscript. This is enumerated in a point-by-point response letter to the reviewers showing specific changes in the revised submission:

Reviewer #1: 

• The experiment in figure 7B should be repeated with a control group for α7345-348a that was not exposed to choline to show that transfection with α7345-348a does not alter proliferation by itself.

RESPONSE: We thank the reviewer for this comment. In fact, the experiment was conducted as a control however it was not included in the original submission. Therefore, the revised manuscript shows data on mutant receptor subunit transfection within Figure 7 and corresponding text in Results page 8 (last sentences of the section) .

• The results in Figure 4 are not mentioned in the abstract or the discussion. It would be good if the authors discuss how the increase in α7 nAChR expression might affect the proliferation, motility, and/or calcium signaling of cancer cells. Additionally, the authors did not show if these effects could be blocked by MLA or another antagonist, or whether this upregulation is related to g-protein mediated signaling via α7 nAChRs. Further experiments to answer these questions would help to tie these results into the rest of the paper.

RESPONSE: We thank the reviewer for pointing this out and we have now incorporated a discussion of this on Page 9 (end of the second paragraph). As stated, alteration in nicotinic receptor expression at the cell surface is likely an important driver of calcium and other signaling responses that can support cancer cell growth, and the experiments suggested by the reviewer are clearly important. However, due to limitation in experimental resources, we are not able to perform these experiments currently. Furthermore, we would like to suggest that studies on the mechanisms that participate in the regulation of nAChR at the cell surface, fall outside of the scope of the current manuscript. We now mention this as an important direction in future study in the revised manuscript in the Discussion on Page 9 (end of the second paragraph).

• YM 254890 is mentioned in the abstract as a G protein blocker, but in the results section of the manuscript, the authors never define what specifically YM 254890 is or why it is being used.

RESPONSE: The correction is now included on Page 7, last sentences of the 3rd paragraph within the Results section. The clarification is made that YM254890 is a Gq inhibitor and used to confirm Gq involvement.

• The proliferation assay in Figure 7 plots Cell count x 105, whereas the other proliferation assays measure Net OD. Could the authors please explain why this assay was quantified differently?

RESPONSE: Proliferation assays in the manuscript were performed in the laboratory of Dr. Oz however those presented in Figure 7 were performed in the lab of Dr. Kabbani. The two assays are virtually identical (i.e., using the same cell line and plating and MTT counting protocol). The results are also highly robust and show a comparable effect of choline on MCF-7 cells proliferation when comparing % change in cell count relative to the control group. To clarify this in the revised manuscript a newly added Figure 7 C shows cell count results as “% control (vehicle)” as done in earlier figures.

• There are two “B” panels in figure 5 

Response: This is now corrected. 

• The deltaF/F scales are not consistent across figures. For example, in Figure 5, A and C plot deltaF/F (x100), whereas panel B1 writes out the 100s, and the second panel B just has deltaF/F without the x100 clarification. Similarly, in Figure 7, panel A, the first graph is x100, while the second graph is not. Scale bar should be defined in Figure 7B.

Response: These figures are significantly revised to correct for these errors. 

• Scale bars should be added to Figure 3A and Figure 4A

Response: The scale bars are now added into Figs. 3 and 4.

• In the figure legend for Figure 6 it is noted that the “Asterix denotes significant difference from control,” whereas the figure seems to indicate that the Asterix represents significant difference from the choline treated group. Could the authors clarify what comparisons are being measured in this figure, and what is significantly different from what?

Response: Thank you for pointing out this error. We have revised the legends for figures 2, 6, and 7 with the new legends clearly stating which groups are used for statistical comparison.

• The figure legend for Figure 2 states “Asterix denoted significant difference from control” but the figure seems to indicate that significance is measured as difference from the choline group.

Response: This is now corrected.

• Figure 7 has only 2 panels, A and B, but the figure legend has A, B, and C.

Response: This is now corrected. 

• Figure 7 states “statistical significance relative to the control or α7 transfected group.” This is confusing. Could the authors clarify what comparisons are being made and what significance is being denoted. Does this mean that the α7345-348a group is significantly different from both the control and α7 group? If so, this should be denoted by two different symbols.

In the text, the mutant α7 is called “α7345-348a” whereas in the figure legend and figure it is “α7345-8a”, this nomenclature should be consistent throughout the paper

Response: This is now corrected. 

• In the figure legend for Figure 5, the definition of a double Asterix is given, but there is no double Asterix in Figure 5

Response: This is now corrected. 

Reviewer #2: The paper by Oz et al, shows that in MCF-7 cells activation by choline of α7 receptor increases

proliferation, motility, and calcium signaling and this is partially blocked by the application of G protein blocker, Gαi/o inhibitor and by the presence of a G protein binding dominant negative α7 subunit (α7345-348A). The work is interesting and well done, and it demonstrated that the α7 subtype is involved in mediating choline induced proliferation, motility, and calcium signalling. 

• However the effect of choline was blocked using antagonists that act on both α7 and α9 subtypes. Why the authors did not used the α7 selective antagonist AR (Whitaker et al. 2007) and the α9 selective antagonist RGIA4 (Romero et al 2017) ? This certainly will help in clarifying the subtypes involved and their contribution to the effects report.

Response: α-Conotoxin RgIA has higher potency for α9/α10 than α7 nAChRs (Ellison et al., 2008), and was thus chosen in complementary experiments to α bungarotoxin. The use of additional pharmacological ligands with greater selectivity, as suggested by the reviewer, is definitely important and to that end this point is now stated in the Discussion on Page 10, last two sentences.

Minor points: In the abstract it is written "Nicotine activates various nicotinic acetylcholine receptors (nAChRs)" and in the introduction it is written"Mammalian nAChRs assemble from α (α1–α6) and β (β2–β4) subunit combinations forming homo- or hetero-pentameric channels [9, 10].

• Nicotine does not activates a1 containing receptors and nicotine activates various nAChRs

that assemble from α (α2–α7, α9, α10) and β (β2–β4) subunit combinations.

• Fig 7: Please add C to the figure

Response: We thank the reviewer for pointing these mistakes and have corrected them within the revised manuscript.

---

## [Decision Letter · Decision Letter 1]

12 Jul 2023

α7 nicotinic acetylcholine receptor interaction with G proteins in breast cancer cell proliferation, motility, and calcium signaling

PONE-D-23-15038R1

Dear Dr. Kabbani,

We’re pleased to inform you that your manuscript has been judged scientifically suitable for publication and will be formally accepted for publication once it meets all outstanding technical requirements.

Kind regards,

Israel Silman

Academic Editor

PLOS ONE

Additional Editor Comments (optional):

Reviewers' comments:

Reviewer's Responses to Questions

**Comments to the Author**

1. If the authors have adequately addressed your comments raised in a previous round of review and you feel that this manuscript is now acceptable for publication, you may indicate that here to bypass the “Comments to the Author” section, enter your conflict of interest statement in the “Confidential to Editor” section, and submit your "Accept" recommendation.

Reviewer #1: All comments have been addressed

2. Is the manuscript technically sound, and do the data support the conclusions?

Reviewer #1: (No Response)

3. Has the statistical analysis been performed appropriately and rigorously? 

Reviewer #1: (No Response)

4. Have the authors made all data underlying the findings in their manuscript fully available?

Reviewer #1: (No Response)

5. Is the manuscript presented in an intelligible fashion and written in standard English?

Reviewer #1: (No Response)

6. Review Comments to the Author

Reviewer #1: (No Response)

7. PLOS authors have the option to publish the peer review history of their article (what does this mean?). If published, this will include your full peer review and any attached files.

Reviewer #1: No

---

## [Editor Report · Acceptance letter]

17 Jul 2023

PONE-D-23-15038R1 

α7 nicotinic acetylcholine receptor interaction with G proteins in breast cancer cell proliferation, motility, and calcium signaling 

Dear Dr. Kabbani:

I'm pleased to inform you that your manuscript has been deemed suitable for publication in PLOS ONE. Congratulations! Your manuscript is now with our production department. 

Kind regards, 

on behalf of

Prof. Israel Silman 

Academic Editor

PLOS ONE